# Prescribed-Time Command Filtered Control for Uncertain Nonlinear Systems

1st Yilin Chen
*College of Control Science and Engineering*
*Bohai University*
Jinzhou, China
2022008001@qymail.bhu.edu.cn

*Abstract*—This paper focuses on the prescribed-time (PT) adaptive tracking control scheme for nonlinear systems. A time-varying scaling transformation function (STF) is designed to avoid the singularity issue in implementing the designed controller while releasing the requirement of n-order differentiability of the STF, thus providing a more relaxed criterion for PT stability. Furthermore, a simplified filtering error compensation method is proposed to ensure the convergence of filtering error in prescribed time, thus reducing the impact of filtering error on control performance. All signals in nonlinear systems are proved to meet the PT stability requirement. The effectiveness of the proposed scheme are substantiated through simulation results.

*Index Terms*—Adaptive tracking control, filtering error compensation, nonlinear systems, prescribed-time control.

## I. INTRODUCTION

Many practical systems can be formulated as nonlinear systems. The backstepping technique stands out as a common and robust approach. The backstepping control technology exhibits several significant features, including global stability and asymptotic tracking, which have been extensively studied in [1] and [2]. Nevertheless, existing research primarily focuses on asymptotical stabilization or uniform ultimate boundedness, implying that system stabilization within finite time remains unexplored.

Recently, great efforts have been achieved in finite-time (FT) control for nonlinear systems [3]. Compared to traditional infinite-time stabilization methods, FT stability has advantages such as high convergence accuracy and fast convergence speed. However, the convergence time of FT control is contingent upon the system's initial condition. Addressing this challenge, the concept of fixed-time (FxT) convergence stability was introduced in [4]. Nevertheless, the convergence time boundary derived from stability analysis often diverges from the actual settling time observed in simulations.

To address the aforementioned issues, the concept of predefined-time (PdT) stability was introduced, which has an attractive advantage of being able to preset the upper bound on settling time. In addition, some related achievements have been obtained in [5, 6]. However, the existing PdT control schemes can only determine the upper bound of the convergence time. Song *et al.* [7] systematically introduced the prescribed-time (PT) control method, employing a scaling transformation function (STF) that diverges towards being unbounded as time approaches the user-defined terminal time. Although there have been a lot of efforts devoted to the PT control, the obtained result mentioned above are relatively conservative. The STFs require n-order differentiability, which lead to enhanced conservatism in the controller design process. Moreover, the PT stability within the backstepping framework causes the "complexity explosion" problem.

Inspired by the aforementioned discussions, an adaptive prescribed-time control method is proposed for nonlinear systems based on a first order sliding mode differentiator. The main contributions are concluded as follows:

1) The filtering technique and the PT control technique are combined to release the limitations of the n-order differentiability of STFs.
2) In this paper, a simplified command filtering error compensation strategy is proposed, which precisely sets the convergence time of filtering errors without the need for scaling.

## II. PRELIMINARIES

### A. Problem Formulation

In this paper, we consider a class of strict-feedback systems with uncertainties, which is modeled by

$$\begin{cases} \dot{x}_i = x_{i+1} + f_i(\bar{x}_i), & i = 1, \ldots, n-1 \\ \dot{x}_n = u + f_n(\bar{x}_n), \\ y = x_1, \end{cases} \tag{1}$$

where $x_i \in \mathbb{R}$ represents the state with $\bar{x}_i = [x_1, x_2, \ldots, x_i]^T$ being the state vector, $u$ is the control input signal and $y$ denotes the output, and $f_i(\bar{x}_i)$ is the uncertain smooth nonlinear function.

### B. Practical Prescribed-Time Stable

*Definition 1:* For the following system:

$$\dot{x} = f(x, t) \tag{2}$$

where $f$ is piecewise continuous in $t$ and locally Lipschitz in $x$, and $x = 0$ is the equilibrium point. Then, system (2) exhibits PT global uniform asymptotic stability over time $T$ if there exists a function $\mu : [0, T) \to R_+$ that tends to infinity as $t$ approaches $T$, and a class $KL$ function $\Re$, such that the norm

of $x(t)$ is bounded by $\Re(\|x(0)\|, (t))$ for all $t$ in $[0, T)$, where $T$ is a finite number determined during the design phase.

We put forward an improved time-varying STF defined on the whole time interval as follows:

$$\beta(t) = \begin{cases} \dfrac{T}{T - t + \varepsilon}, & t \in [0, T) \\ \dfrac{T}{\varepsilon}, & t \in [T, +\infty) \end{cases} \tag{3}$$

where $T$ represents any finite time duration permissible within physical constraints set by the user, and $\varepsilon$ denotes a small constant.

*Lemma 1:* Given a radially unbounded positive definite LF $V(t)$

$$\dot{V}(t) \leq -c_1 V(t) - \frac{\dot{\beta}(t)}{\beta(t)} V(t) + b \tag{4}$$

*C. Fuzzy Logic Systems (FLSs)*

The FLSs are used to estimate $f(x)$ over a compact set $\Omega$ as

$$f(x) = W^T \delta(x) + \epsilon(x), \forall x \in \Omega$$

where $W$ is the artificial constant vector given as $W = \arg\min_{W} \left[ \sup_t \left| f(x) - \hat{f}(x) \right| \right]$. $\epsilon(x)$ is the approximation error bounded by a positive constant $\kappa$, that is, $|\epsilon(x)| \leq \kappa$. Moreover, $\delta(x)$ is the fuzzy basis vector denoted as

$$\delta(x) = \frac{\prod\limits_{\tau=1}^{\tilde{N}} \mu_{F_\tau}(x_\tau)}{\sum\limits_{\tau=1}^{r} [\prod\limits_{\tau=1}^{\tilde{N}} \mu_{F_\tau}(x_\tau)]},$$

where $\mu_{F_\tau}$ is the membership degree of $x_\tau$, and $N$ is the number of total fuzzy rules.

## III. ADAPTIVE PRESCRIBED-TIME CONTROLLER DESIGN

Prior to finalizing the deduction and formulating the prescribed-time controller, the subsequent coordinate transformation will be delineated as

$$\begin{aligned} z_1 &= y - y_r, \\ z_i &= x_i - \alpha_{i-1}, i = 2, \ldots, n \end{aligned} \tag{5}$$

where $\alpha_{i,n-1}$ is the virtual control signal.

According to (1) and (5), $\dot{z}_1$ can be deduced as

$$\begin{aligned} \dot{z}_1 &= \dot{x}_1 - \dot{y}_r \\ &= f_1 + z_2 + \alpha_1 - \dot{y}_r. \end{aligned} \tag{6}$$

**Step 1**: Construct the Lyapunov function as

$$V_1 = \frac{1}{2} z_1^2 + \frac{1}{2} \theta_1^2, \tag{7}$$

where $\theta_1$ is defined as $\theta_1 = \|W_1\|^2$. $W_1$ is the artificial constant vector which will be employed later.

Then, the time derivative of $V_1$ represents

$$\dot{V}_1 = z_1(f_1 + z_2 + \alpha_1 - \dot{y}_r) - \tilde{\theta}_1 \dot{\hat{\theta}}_1, \tag{8}$$

where $\hat{\theta}_1$ is the estimation of $\theta_1$, and the error between $\theta_1$ and $\hat{\theta}_1$ is defined as $\tilde{\theta}_1 = \theta_1 - \hat{\theta}_1$. $f_1$ is the unknown term, which can be estimated by FLSs to arbitrary accuracy $\kappa_1$, namely,

$$f_1 = W_1^T \delta_1 + \epsilon_1, |\epsilon_1| \leq \kappa_1 \tag{9}$$

where $W_1$ is the artificial constant vector, $\delta_1$ is the fuzzy basis function, and $\epsilon_1$ is the approximation error.

Then, one has

$$\dot{V}_1 \leq z_1 z_2 + z_1(\alpha_1 - \dot{y}_r) - \tilde{\theta}_1 \dot{\hat{\theta}}_1 + z_1 \lambda_1 + \frac{1 + z_1^2 + \kappa_1^2}{2}. \tag{10}$$

The virtual control signal $\alpha_1$ is designed as

$$\alpha_1 = (-\frac{1}{2} c_1 - \frac{\dot{\beta}}{2\beta}) z_1 - \frac{1}{2} z_1 + \dot{y}_r - \hat{\lambda}_1, \tag{11}$$

where $\hat{\lambda}_1$ is the estimation of $\lambda_1$, and the error between $\lambda_1$ and $\hat{\lambda}_1$ is defined as $\tilde{\lambda}_1 = \lambda_1 - \hat{\lambda}_1$.

The adaptive law is designed as

$$\dot{\hat{\theta}}_1 = (-\frac{1}{2} c_1 - \frac{\dot{\beta}}{2\beta}) \hat{\theta}_1 + \frac{1}{2} z_1^2 \delta_1^T \delta_1. \tag{12}$$

Then, we have

$$\dot{V}_1 \leq z_1 z_2 + (-c_1 - \frac{\dot{\beta}}{\beta}) \frac{1}{2} z_1^2 - (c_1 + \frac{\dot{\beta}}{\beta}) \frac{1}{2} \tilde{\theta}_1^2 + \varrho_1, \tag{13}$$

where $\varrho_1 = \frac{1}{2} + \frac{1}{2} \kappa_1^2 + (\frac{1}{2} c_1 + \frac{\dot{\beta}}{2\beta}) \theta_1^2$.

**Step $i(i = 2, \ldots, n-1)$:**

Given the computational complexity of calculations, we employ the first-order sliding mode differentiator to estimate $\dot{\alpha}_{i-1}$.

Hence, we can obtain

$$\dot{\alpha}_{i-1} = p_{i-1} + e_{i-1}, \tag{14}$$

where $e_{i-1}$ is the estimated error of differentiator.

The filtering error compensation mechanism is devised as

$$e_{i-1} = \hat{e}_{i-1} + \tilde{e}_{i-1}, \tag{15}$$

where $\hat{e}_{i-1}$ represents the adaptive law for approximating the filtering error $e_{i-1}$, and $\tilde{e}_{i-1}$ is the error between $e_{i-1}$ and $\hat{e}_{i-1}$.

The Lyapunov function candidate is chosen as

$$V_i = V_{i-1} + \frac{1}{2} z_i^2 + \frac{1}{2} \theta_i^2 \tag{16}$$

Differentiating $V_i$, we can get

$$\dot{V}_i = \dot{V}_{i-1} + z_i(f_i + z_{i+1} + \alpha_i - p_{i-1} - e_{i-1}) - \tilde{\theta}_i \dot{\hat{\theta}}_i \tag{17}$$

Then, (17) can be re-expressed as

$$\begin{aligned} \dot{V}_i = &\dot{V}_{i-1} + z_i(\alpha_i - p_{i-1} - e_{i-1}) - \tilde{\theta}_i \dot{\hat{\theta}}_i \\ &+ z_i \lambda_i + \frac{1 + z_i^2 + \kappa_i^2}{2} + z_i z_{i+1}. \end{aligned} \tag{18}$$

The virtual control signal $\alpha_i$ is designed as

$$\alpha_i = (-\frac{1}{2}c_1 - \frac{\dot{\beta}}{2\beta})z_i - z_{i-1} + p_{i-1} \\ + \hat{e}_{i-1} - \frac{1}{2}z_i - \hat{\lambda}_i, \tag{19}$$

The adaptive law is designed as

$$\dot{\hat{\theta}}_i = (-\frac{1}{2}c_1 - \frac{\dot{\beta}}{2\beta})\hat{\theta}_i + \frac{1}{2}z_i^2 \delta_i^T \delta_i. \tag{20}$$

Substituting (19) and (20) into (18), one has

$$\dot{V}_i \le -\sum_{j=1}^{i}(c_1 + \frac{\dot{\beta}}{\beta})\frac{1}{2}z_j^2 - \sum_{j=1}^{i}(c_1 + \frac{\dot{\beta}}{\beta})\frac{1}{2}\tilde{\theta}_j^2 \\ \sum_{j=2}^{i} z_j \tilde{e}_{j-1} + z_i z_{i+1} + \varrho_i, \tag{21}$$

where $\varrho_i = \varrho_{i-1} + \frac{1}{2} + \frac{1}{2}\kappa_i^2 + (\frac{1}{2}c_1 + \frac{\dot{\beta}}{2\beta})\theta_i^2$.

**Step $n$**: Select the Lyapunov function as

$$V_n = V_{n-1} + \frac{1}{2}z_n^2 + \frac{1}{2}\tilde{\theta}_n^2. \tag{22}$$

In the final step, the actual controller $u$ is designed as follows:

$$u = (-\frac{1}{2}c_1 - \frac{\dot{\beta}}{2\beta})z_n - z_{n-1} + p_{n-1} \\ + \hat{e}_{n-1} - \frac{1}{2}z_n - \hat{\lambda}_n, \tag{23}$$

The adaptive law is designed as

$$\dot{\hat{\theta}}_n = (-\frac{1}{2}c_1 - \frac{\dot{\beta}}{2\beta})\hat{\theta}_n + \frac{1}{2}z_n^2 \delta_n^T \delta_n. \tag{24}$$

Then, we can get

$$\dot{V}_n \le -\sum_{j=1}^{n}(c_1 + \frac{\dot{\beta}}{\beta})\frac{1}{2}z_j^2 - \sum_{j=1}^{i}(c_1 + \frac{\dot{\beta}}{\beta})\frac{1}{2}\tilde{\theta}_j^2 \\ -\sum_{j=2}^{n} z_j \tilde{e}_{j-1} + \varrho_n, \tag{25}$$

where $\varrho_n = \varrho_{n-1} + \frac{1}{2} + \frac{1}{2}\kappa_n^2 + (\frac{1}{2}c_1 + \frac{\dot{\beta}}{2\beta})\theta_n^2$.

*Theorem 1:* For the nonlinear systems with uncertainties (1), under the first order differentiator (15), the filtering error compensation mechanism (14), the controller (23), the adaptive law (20) and the adaptive filtering update law (27), it can guarantee that the tracking error $z_1$ converges to a small neighborhood of origin and all signals are prescribed-time bounded.

*Proof:* For the compensating mechanism (15), select the Lyapunov function as

$$V_c = \frac{1}{2}\sum_{j=2}^{n}\tilde{e}_{j-1}^2 \tag{26}$$

To compensate for the impact of filtering errors, the adaptive filtering update law is designed as

$$\dot{\hat{e}}_{j-1} = -z_j - (c_1 + \frac{\dot{\beta}}{\beta})\hat{e}_{j-1}. \tag{27}$$

Select $V = V_n + V_c$ as the Lyapunov function candidate of the nonlinear systems (1). Differentiating $V$, $\dot{V}$ can be expressed as

$$V \le - c_1 V - \frac{\dot{\beta}}{\beta}V + b \tag{28}$$

From Lemma 1, (28) shows that the tracking error and the filtering error will converge in the prescribed time. This ends the proof.

## IV. SIMULATION RESULTS

In this section, two examples are given to demonstrate the validity of prescribed-time command filtering adaptive fuzzy controller.

Example 1: Consider the uncertain nonlinear systems as follows:

$$\begin{cases} \dot{x}_1 = x_2 + f_1(\bar{x}_1) \\ \dot{x}_2 = u + + f_2(\bar{x}_2) \\ y = x_1 \end{cases} \tag{29}$$

where $f_1(\bar{x}_1) = x_1 \exp(-0.5x_1)$ and $f_2(\bar{x}_2) = -2x_1 - x_2$. The reference signal $y_r$ is defined as $y_r = 0.1\sin(2t)$.

In the simulation, the designed parameters are chosen as $c_1 = c_2 = 20$ and $\varepsilon = 0.1$. The initial states are chosen as $x_1(0) = -0.3$ and $x_2(0) = -0.1$. The adaptive parameters are initialized to be $\hat{\theta}_1(0) = 0.2$, $\hat{\theta}_2(0) = 0.4$ and $\hat{e}_1(0) = 0.3$, respectively.

The output signal and reference signal trajectories are depicted in Fig. 1. Fig. 2 illustrates the tracking error. From Fig 3, it can be seen that the impact of filtering error has been reduced within the prescribed time.

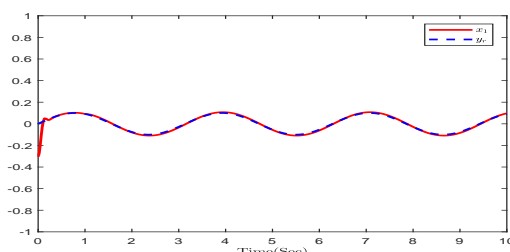

Fig. 1. PT tracking control trajectory with $T = 1s$.

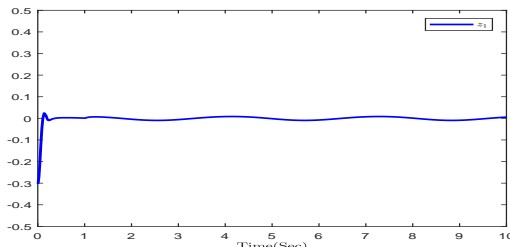

Fig. 2. The curve of the tracking error $z_1$ with $T = 1s$.

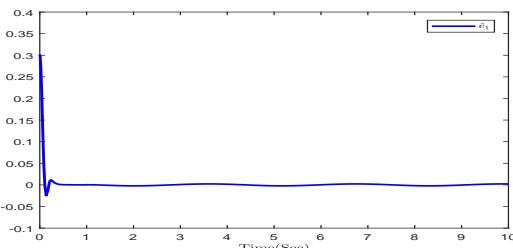

Fig. 3. The curve of filtering error compensation adaptive parameter $\hat{e}_1$ with $T = 1s$.

## V. CONCLUSION

In this paper, the FLS-based PT adaptive tracking control problem has been considered for nonlinear systems. A simplified filtering error compensation mechanism has been proposed to make filtering errors converge in the prescribed time. Finally, the designed PT command filtering controller ensures that all signals are PT bounded.

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
