# OpenReview forum: "Prescribed-Time Command Filtered Control for Uncertain Nonlinear Systems"
_IEEE.org/ICIST/2024/Conference — IEEE ICIST 2024 Conference Submission_

### Official Review · Reviewer_zvFz · 2024-08-20
**Prescribed-Time Command Filtered Control for Uncertain Nonlinear Systems**

**Rating:** 2
**Confidence:** 5

**Review:**

This paper focuses on the prescribed-time adaptive tracking control scheme for nonlinear systems. A time-varying scaling transformation function is designed to avoid the singularity issue in implementing the designed controller. However, the contributions of this work are unclear. The presented work seems to be a combination of some existing results. After careful evaluation, I regretfully believe that this paper cannot be accepted.

---

### Official Review · Reviewer_Mrpw · 2024-08-24
**The overall workload of the paper is insufficient.**

**Rating:** 2
**Confidence:** 4

**Review:**

1. The innovation of this paper is not sufficient.
2. The paper falls short in terms of the comprehensiveness and rigor required for a scholarly work, as it lacks extensive research and empirical evidence.

---

### Official Review · Reviewer_zpCc · 2024-08-24
**The paper proposes a PT adaptive tracking control method for nonlinear systems with a new STF that lowers the smoothness requirement, simplifying control implementation. It also introduces a filtering error compensation strategy to improve control performance. Despite these contributions, the paper's novelty is questionable, and the lack of comparative tests limits the demonstration of its uniqueness and effectiveness.**

**Rating:** 3
**Confidence:** 5

**Review:**

This paper is about a special way to control systems that don't always behave the same way (nonlinear systems). It talks about a method called "prescribed-time" (PT) adaptive tracking control, which helps to make sure that the system follows a certain path at the right time.The paper introduces a new function called the "time-varying scaling transformation function" (STF). This function is used to fix a problem where the control method might not work well if the STF isn't smooth enough. By using this new STF, the paper says we don't need the STF to be super smooth, which makes it easier to use.Also, the paper suggests a new way to handle errors that come from something called "filtering." This new method makes sure that these errors get smaller in the time that we want (prescribed time), which helps the control system work better.In the end, the paper shows that all the parts of the nonlinear systems work well with this new method. It also shows through computer simulations that the new control scheme works as expected.

    However, the innovation of this paper is not novel enough, and the work based on the backstepping method and the proposed simplified command filtering error compensation strategy do not reflect the uniqueness of the work they have done.
    The paper does not have enough work and lacks comparative tests.

---

### Decision · Program_Chairs · 2024-09-08

Reject